# Study on the influence of topography on wind shear-numerical simulation based on WRF-CALMET

Xingyu Wang[1], Yuhong Lei[1], Baolong Shi[1], Zhiyi Wang[1], Xu Li[1], Jinyan Wang[1]

[1]Key Laboratory of Climate Resource Development and Disaster Prevention in Gansu Province, College of Atmospheric Sciences, Lanzhou University, Lanzhou 730000, China

**Correspondence**: Jinyan Wang (wangjny@lzu.edu.cn)

## Abstract

This study focuses on the critical issue of low-altitude wind shear, vital for aircraft safety during takeoff and landing. Using the WRF-CALMET model, we assess the impact of topography on low-level wind shear at Zhongchuan Airport. CALMET outperforms WRF, showing improved simulation accuracy.CALMET's simulation highlights diurnal variations in vertical wind shear, especially pronounced from 13:00 to 24:00. Notably, CALMET indicates 1-2 hazard levels higher wind shear for aircraft operations compared to WRF in a significant area.Terrain sensitivity experiments reveal CALMET's responsiveness to terrain changes during high wind shear periods, with reduced impact at higher altitudes. CALMET's incorporation of kinematic terrain influences, blocking effects, slope flow, and strengthened diversion of near-surface airflow on complex terrain contribute to these findings.This study confirms the efficacy of CALMET in simulating low-altitude wind shear, emphasizing its superiority in capturing terrain influences and reducing the aviation safety threat posed by low-altitude wind shear.

**Keywords** — wind shear; wind field-numerical simulation; airport; CALMET; aeronautical meteorology; topographic effect

## 1. Introduction

According to the definition of the International Civil Aviation Organization (ICAO), low-level windshear refers to the sharp change of spatial wind speed or direction within a 600-meter altitude range. Wind shear includes both vertical and horizontal components and typically occurs near fronts, coastlines and the surface. In the process of taking off and landing, low-level wind shear will affect the airspeed of the aircraft, causing great risks and even terrible accidents in serious cases (Evans and Turnbull, 1989). In June 1975, a Boeing 727 aircraft crashed during its landing at Kennedy Airport due to encountering low-level wind shear, resulting in 113 fatalities and 11 injuries (Fujita and Caracena, 1997); In June 2000, a Wuhan Airlines aircraft crashed during landing, also due to encountering low-level wind shear. In 2017, a

New Zealand Airlines A320-200 aircraft experienced low-level wind shear during
landing, resulting in severe damage to the aircraft and significant economic losses.
Therefore, accurate simulation and prediction of low-level wind shear, especially on
complex terrain, is of great significance for ensuring the safety of aircraft takeoffs and
landings at airports.
However, achieving accurate predictions remains a primary challenge faced by
numerical weather forecasting models (Colman et al., 2012). Low-level wind shear is
influenced by multiscale weather systems and characterized by small temporal and
spatial scales, high intensity, and sudden occurrences, thus making it difficult to
detect, study and predict. In simulating actual wind fields, simple characteristics are
insufficient; the wind field structure around the airport must be included. There are
three main methods for calculating wind shear in model wind fields (Zhang and Jia,
2022):1.Using meteorological radar networks and various monitoring networks
around airports, differential methods are employed to collect measured data,
recording wind speed, and wind direction in a grid format. However, these
measurements are scattered and small, insufficient to capture the essential
characteristics and dynamic development of low-level wind shear, and do not vary
with meteorological conditions.2.The second type of wind shear model is common in
engineering and consists of simple models. These typically comprise some physical
concepts, represented through simple mathematical fitting and basic fluid dynamics
solutions. They only reflect essential features of the shear wind field without fully
capturing the true wind field characteristics (Li et al., 2016).3.The third type of wind
shear model is based on atmospheric dynamics and physical equations, solved
directly by large computers. Among these methods, the third not only simulates the
real wind shear in the wind field but also provides other useful physical quantities
(e.g., temperature, water content, and radar reflectivity), revealing the formation
process, causes, and development of wind shear. Many studies have utilized
numerical models to simulate low-level wind shear.
Boilley used the non-hydrostatic Meso-NH model to simulate two different wind
shear events in the complex terrain around Nice Côte d'Azur Airport. They
successfully predicted vertical wind shear and local turbulence; however, due to the
model resolution limitation (500m), the study did not accurately predict the time and
location of low-level wind shear. Consequently, subsequent wind shear studies have
continuously improved spatial resolution. The Weather Research and Forecasting
(WRF) model, designed for high-resolution mesoscale weather forecasting, simulates
airflow under realistic atmospheric conditions. However, due to the grid resolution
of WRF being greater than 1 km, it struggles to simulate the small-scale airflow
movements in complex terrain. Hong Kong International Airport previously
attempted to predict wind shear using the WRF model, affirming its capability to
forecast wind shear induced by terrain changes several hours in advance and studied
the model's performance under non-temperature inversion conditions, it reproduced
wind shear characteristics well. However, providing precise warnings for the airport
proved challenging (Chan and Hon, 2016).Building on this, Hong Kong International
Airport conducted further research: using a 200m resolution numerical weather
prediction model, AVM, designed for fine short-term weather forecasting based on
WRF3.4.1 and during the whole research period, the results consistent with the
model forecast were observed on both runways (Hon, 2020). Since then, Hong Kong
International Airport improved the WRF-based coupled model, utilizing the WRF-LES
coupled model to capture many wind characteristics and micro-scale airflow within
the airport, accurately reproducing real wind direction changes (Chen et al., 2022).
These studies demonstrate the effectiveness of numerical models in simulating low-
level wind shear in airport regions, with higher resolution models providing better
simulation results. The series of studies conducted at Hong Kong International
Airport suggests that improving models based on the WRF model or coupling it with
other models is a promising approach for studying low-level wind shear. In previous
studies, the WRF/CALMET coupled model has never been used to study low-level
wind shear in airport regions. This study uses this model, significantly improving
simulation resolution and leveraging CALMET's advantages in wind field calculations,
providing a new method for numerical simulation of low-level wind shear in airport
areas.

Lanzhou Zhongchuan International Airport stands as one of the largest aviation
hubs in Northwest China, situated in the southeastern part of the Qinwangchuan
alluvial-fan basin, surrounded by mountains on all sides. The region is known for
frequent wind shear occurrences, a phenomenon that has become increasingly
common at Lanzhou Zhongchuan Airport due to the rapid growth in the number of
flights. Most wind shear events occur during spring and summer, particularly in May,
June, and July (Li et al., 2020). Statistical reports on wind shear at Lanzhou
Zhongchuan Airport indicate that the majority of incidents occur in the afternoon and
evening. This trend is attributed to the downward momentum in the afternoon,
enhanced convective activity from increased ground heating, and higher wind speeds.
Severe convective weather is more likely to occur in the late afternoon to evening,
contributing to a higher frequency of reported low-level wind shear events.
Conversely, fewer flights operate during the night, accompanied by reduced
convective weather, resulting in relatively fewer reports of aircraft encountering low-
level wind shear (Dang et al., 2013). In May 2016, Zhongchuan Airport installed
coherent Doppler lidar near the runway to study the characteristics of low-level wind
shear and provide warnings  (Li et al., 2020). Numerical simulation studies on wind
shear at Zhongchuan Airport have been ongoing. Jiang L. et al. selected a 6 km×6 km
area near the runway at Zhongchuan Airport to establish a digital elevation model of
the terrain. They used FLUENT software for numerical simulation, solving iterative
calculations to obtain the distribution characteristics of wind speed and pressure in
the simulated area (Jiang et al., 2018). However, FLUENT, being a Computational
Fluid Dynamics (CFD) simulation software widely used in engineering, science, and
research fields, only considers the local turbulence of terrain and buildings on the
flow field. It does not account for factors such as gravity and heat exchange in real
atmospheric conditions. Therefore, relying solely on FLUENT for simulating and
warning wind shear at Zhongchuan Airport has its limitations. Improvements in
simulating low-level wind shear still require enhancements built upon numerical
weather forecasting models.
In both domestic and international research, the CALMET model is frequently
employed to downscale WRF, providing a finer representation of microscale terrain
structures. Particularly in weak wind conditions, the CALMET downscaling coupling
model outperforms WRF in simulating near-surface wind directions(Zhang et al.,
2020). The WRF/CALMET coupled system demonstrates satisfactory performance in
various challenging scenarios, including the complex terrain of the Qinghai-Tibet
Plateau(Liao et al., 2021) and the intense weather system of Super Typhoon Meranti
(2016)(Tang et al., 2021). Up to now, no one has used WRF/CALMET coupling system
to simulate and test the occurrence of low-altitude wind shear. Therefore, this study
leveraged the dynamic downscaling effect of the CALMET model on local micro-
terrain to achieve high-resolution wind shear simulations with relatively low
computational requirements within a small area. Additionally, we conducted
controlled variable experiments by modifying the original terrain. This approach has
not been attempted in studies investigating terrain-induced wind shear at other
airports. It provides an improved method for simulating low-level wind shear within
the WRF model.

## 2.Mode, Data, Method and Experimental Setup

2.1 Models and Experimental Setup
In this study, the WRF model (version 4.2) was employed to simulate a severe
convective weather event occurring in the vicinity of Zhongchuan Airport over a
duration of 96 hours, starting from July 2, 2022, at 0000 UTC. The simulated wind
field results were then downscaled to 100 meters through coupling with the CALMET
model. The model utilized a three-layer, two-way nested domain configuration
(Figure 1a), with horizontal grid spacings of 9 km, 3 km, and 1 km. In the vertical
direction, there were 39 complete Eta layers from the surface to 0 hPa. The physical
schemes employed by WRF are detailed in Table 1.

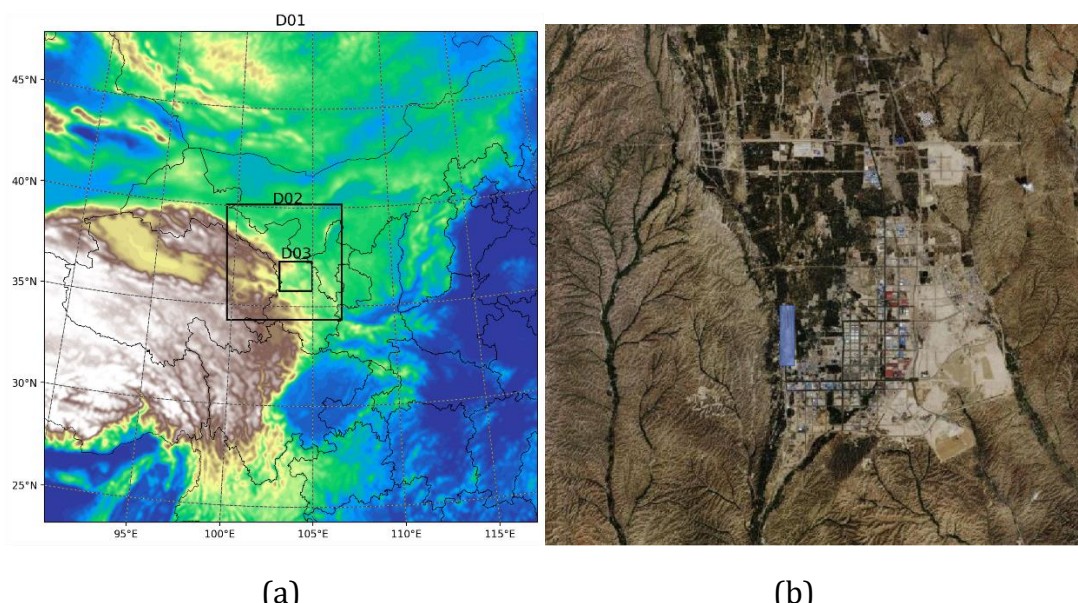

(a)                                    (b)

Figure 1. Three-layer Nested Domains of the WRF Model (a) and Simulation Area of
the CALMET Model © Google Maps(b), with the Zhongchuan Airport Highlighted in
Blue
Table 1. Model Configuration

| Physical Scheme | WRF Option |
|---|---|
| Microphysics | Thompson graupel scheme (2-moment scheme in V3.1) |
| Cumulus parameterization | Tiedtke scheme |
| Longwave radiation | RRTMG |
| Shortwave radiation | RRTMG |
| Surface layer | Monin-Obukhov (Janjic Eta) scheme |
| Land surface | Noah |
| Boundary layer | MYJ |

The diagnostic model utilized in this research is the CALMET model (version 6.5),
which constitutes the meteorological component of the California Puff Dispersion
Model (Scire J S.2000). In the configuration of this study, the initial guessed wind field
is obtained from the grid wind field generated by the innermost domain of WRF, with
a horizontal grid spacing of 1 km (D3 in Figure 1(a)). Since no objective analysis
procedure is employed, we only pay attention to the first step wind field.  The
coverage area of the CALMET model encompasses Zhongchuan Airport and its
surrounding 38km×38km region (Figure 1(b)), with a horizontal resolution of 100m.
The vertical layers are set to 10 height levels within 600 m from the ground (the
height range influenced by low-level wind shear).
Terrain Sensitivity Experiments for Demonstrating the Impact of Terrain on
Wind Shear Simulation in CALMET:
(1) CALMET: CALMET model configured with default settings as described earlier.
(2) CALEMT_FLAT: Modification in the TERREL terrain processing module where
the elevation of all grid points is adjusted to 2000 meters. This adjustment facilitates
CALMET simulation on a flat underlying surface.
(3) CALEMT_RAISE: Modification in the TERREL terrain processing module
where the elevation of grid points with an altitude exceeding 2050 meters is
increased by 1.5 times. This modification enables CALMET to simulate wind shear
over a more rugged terrain.
These terrain sensitivity experiments are designed to showcase how variations
in terrain impact wind shear simulation within CALMET. The CALMET_FLAT
experiment simulates wind shear on a flat surface, while the CALMET_RAISE
experiment explores wind shear simulation over steeper terrain. The comparison of
results from these experiments with the default CALMET setting will provide insights
into the sensitivity of wind shear simulations to terrain variations.
2.2 data
The terrain data comes from the global 90 m digital elevation data set of Shuttle
Radar Topography (SRTM3 V4.1) of NASA, and the land use data comes from the
global land cover type data with 10m resolution of Pengcheng Laboratory
(https://data-starcloud.pcl.ac.cn/zh) of Tsinghua University in 2017.
The horizontal resolution of the ECMWF Reanalysis v5 (ERA5) dataset is 0.25° ×
0.25°, with a temporal resolution of 1 hour. This dataset is employed as both the initial
input and boundary fields for WRF model. Additionally, this study utilizes ERA5
variables, specifically geopotential height and temperature, for analyzing weather
systems during periods of intense convection.
Observational data for ground-level 10m wind speed at Lanzhou Zhongchuan
Airport are sourced from historical wind speed records provided by the National
Oceanic and Atmospheric Administration (NOAA)
(https://www.ncei.noaa.gov/maps/daily/) with a temporal resolution of 1 hour. The
ground 10m wind speed data of WRF model, CALMET model and ERA5 reanalysis
data are interpolated to the location of Zhongchuan Airport, and compared with the
observed data to verify the performance of the models.
2.3 method
To quantify the differences in 10m wind speed among the experiments, the
following statistical metrics are employed:
Index of agreement (IA):
$$\text{IA} = 1 - \frac{\sum_{i=1}^{N}(P_i - O_i)^2}{\sum_{i=1}^{N}(|P_i - \bar{O}| + |O_i - \bar{O}|)^2}$$ (1)

Root-mean-squared error (RMSE):
$$\text{RMSE} = \sqrt{\frac{1}{n}\sum_{i=1}^{n}(O_i - P_i)^2}.$$ (2)
Mean relative error (MRE):
$$\text{MRE} = \frac{1}{n}\sum_{i=1}^{n}\frac{(P_i - O_i)}{O_i}$$ (3)
Here, $\bar{O}$ and $\bar{P}$ represent the average values of observational and simulated data,
respectively. Each observed value is denoted as $O_i$, and each simulated value is
denoted as $P_i$. Smaller values for MRE and RMSE, and an IA closer to 1.0,indicate
better simulation performance.
Wind shear can be categorized into three types: vertical shear β, meridional
horizontal shear α_1, and zonal horizontal shear α_2. Among these, vertical shear of
horizontal wind has a more significant impact on aircraft takeoff and landing
compared to the other types(Bretschneider et al., 2022). It results in changes in wind
speed and direction as an aircraft moves through different altitudes, which can lead
to drastic changes in airflow during ascent or descent, thereby increasing flight
difficulty, particularly during takeoff and landing(Keohan, 2007; Eggers et al., 2003).

## 3.Result


3.1 Improvement of WRF/CALMET coupling model for simulation of low-level
wind shear.
We evaluated the performance of two models in simulating near-surface wind
speeds, as shown in Figure 2 and Table 2. Both models showed better agreement with
observed data during periods of low wind speeds before convective development
(06:00 on July 3) and after convective cessation (02:00 on July 5). During periods of
intense convection, both models captured wind speed variability. Although both
experiments underestimated or overestimated peak wind speeds on July 3 and July 4,
CALMET slightly outperformed WRF in simulating high wind speeds. Furthermore,
Table 2 indicates that CALMET's Mean Relative Error and Root Mean Squared Error
were lower than those of WRF throughout the entire simulated period, with
improvements of 11.13% and 7.24%, respectively. CALMET's Index of Agreement
was also closer to 1 compared to the WRF experiment, with an improvement of
12.06%. These results demonstrate CALMET's superior overall simulation
performance compared to WRF.

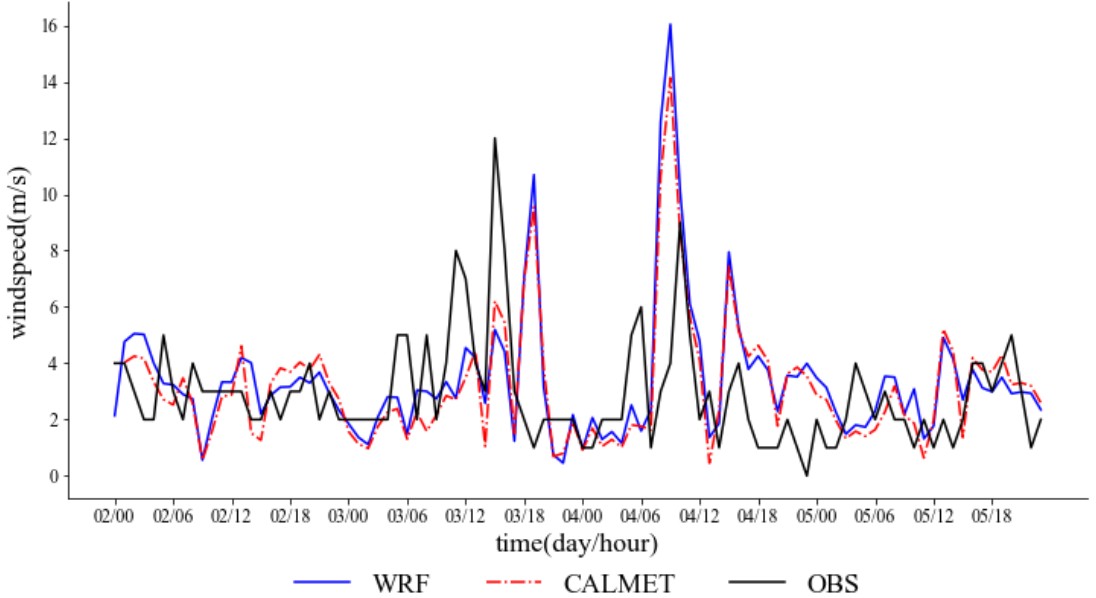


Figure 2 :the time series of 10m surface wind speed for both numerical
simulations and observational data


Table 2:Statistic results of near-surface wind speed simulations in different
experiments averaged.

|  | WRF | CALMET | Improvement (%) |
| --- | --- | --- | --- |
| MRE (%) | 43.255 | 38.425 | 11.13 |
| RMSE(m /s) | 2.713 | 2.517 | 7.24 |
| IA | 0.454 | 0.509 | 12.09 |

At 16:00 on July 3, significant fluctuations in surface wind speeds mark the onset
of convective development (Figure 2). Figure 3 illustrates the distribution of Vertical
Wind Shear (VWS) simulated by both models. In the layer between 10m and 30m
above ground level, CALMET's maximum VWS values, while consistent in location
with WRF's, are notably higher. Terrain analysis reveals CALMET simulates high VWS
values near mountain foothills and western slopes(Figure 8). WRF's high VWS values
primarily occur in mountainous regions. Details for the height layers of 200m-300m
and 500m-600m can be found in the appendix. Overall, both models exhibit
decreasing VWS with increasing height. From the overall distribution of VWS,
CALMET can simulate a wider range of third and fourth level wind shears, which are
associated with severe and extreme turbulence affecting aircraft takeoff and landing.
Furthermore, this capability provides valuable warnings for aircraft operations at
Nakawa Airport.
The atmosphere above and surrounding the mountainous terrain is
characterized by three distinct regions or inclined layers, comprising the thermal
structure undergoing diurnal variations and forming diurnal winds: slope
atmosphere, valley atmosphere, and mountain atmosphere (Zardi and Whiteman,
2013). It is challenging to observe any pure form of diurnal mountain wind system,
as each component interacts with the others. Well-organized thermally driven flows
can be identified over a broad spatial scale, ranging from the dimensions of the largest
mountain ranges to the smallest local topography. Therefore, concerning wind shear
in mountainous and foothill areas, wind shear in mountainous areas tends to be
smaller. When airflow passes through mountain ridges, the lower-level airflow
experiences significant compression. According to the conservation of flux, the
acceleration effect on lower-level airflow exceeds that on upper-level airflow,
resulting in an overall reduction in wind shear. When the acceleration effect on lower-
level airflow is significant while the upper-level acceleration effect is weak or absent,
negative wind shear occurs. Overall, the intensity of low-level wind shear may be
greater near mountain foothills or ridges and lesser in valleys or slopes. Hence, the
regions of maximum wind shear simulated by CALMET near mountain foothills or
ridges are more consistent with reality than those by WRF.

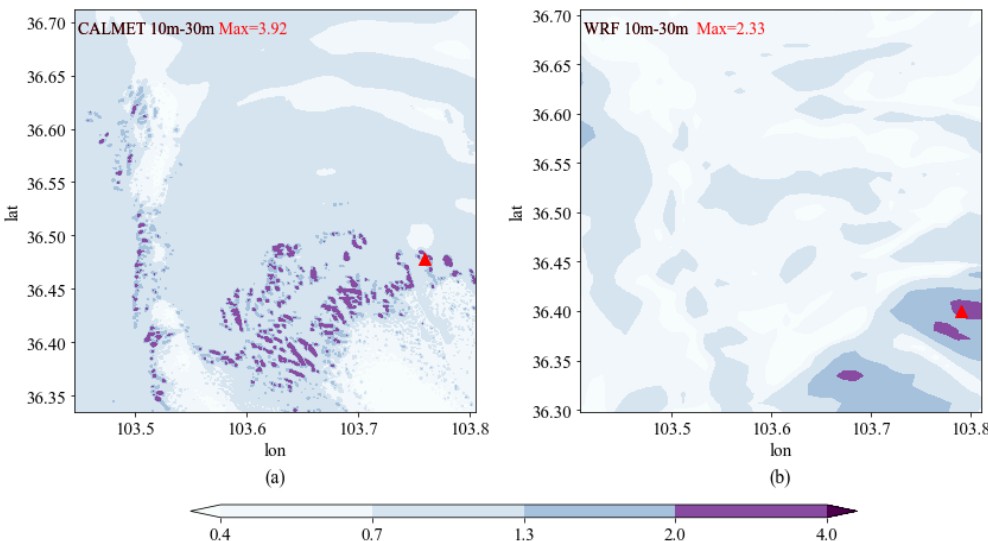

Figure 3: Vertical Wind Shear (VWS) at 16:00 on July 3, 2022, simulated by CALMET (a) and WRF (b) (Unit: m/s/10m). Triangles indicate the locations of maximum values.

Figure 4 presents the time series of maximum VWS simulated by WRF and CALMET. It can be observed that both WRF and CALMET simulations exhibit a clear diurnal pattern in maximum VWS: maximum values are relatively small around dawn and in the morning (1:00 to 12:00), with minimal fluctuations, while they increase significantly in the afternoon and evening (13:00 to 24:00), showing larger variations. However, the maximum values simulated by WRF are generally lower than those by CALMET, with this difference being more pronounced in the afternoon and evening. On July 3rd and 4th, during periods of intense convective activity, CALMET is able to simulate larger fluctuations in maximum VWS compared to normal conditions.

In summary, utilizing CALMET for downscaling WRF output of wind fields provides higher resolution and more precise surface conditions, which are advantageous for simulating mesoscale wind shear. This is primarily manifested in the following aspects: the distribution of VWS in the mid-to-low levels is more significantly influenced by terrain, and VWS decreases more rapidly with increasing altitude; the diurnal variation of maximum VWS within VWS regions follows a clear pattern and can reflect the characteristics of intense convection.

3.2 Impact of Topography on Wind Shear Simulation

Through different terrain configurations, we explored CALMET's detailed terrain impact on low-level wind shear. We found that valley winds affect VWS diurnal variation. Terrain, blocking effects, and slope flow kinematics enhance near-surface airflow diversion, deflection, and ascent over complex terrain, significantly influencing VWS, with the impact decreasing with height.

In the CALMET_FLAT experiment, the increase in maximum VWS during the
afternoon and evening is minimal (Figure 4), with slight fluctuations and values
around 2 m/s/10m, sometimes even lower than WRF. However, good agreement is
observed among the three experiments during the early morning and morning
periods. In CALMET_RAISE, particularly on July 3rd and 4th during intense convective
development, fluctuations in the afternoon and evening are more pronounced
compared to CALMET. However, CALMET_RAISE shows stability similar to CALMET
just before convective development on July 2nd, except for an unusually high value at
09:00 on July 4th, where fluctuations are more pronounced, but numerically close to
CALMET.
In the afternoon and evening, CALMET_FLAT shows a significant decrease in
maximum VWS, while CALEMT_RAISE exhibits more pronounced fluctuations. For
example, at 19:00 on July 3rd (Figure 5(a)-(c)), in the CALMET experiment, the
maximum VWS (3.56 m/s/10m) occurs in the southeastern foothills and valley areas.
In CALMET_FLAT, except for the absence of a high-value area in the southeast, the
distribution is similar to CALMET, with a maximum value of 1.77 m/s/10m in the
central region, which is also a flat valley area in CALMET. In CALMET_RAISE, due to a
sudden 1.5-fold increase in terrain elevation above 2050m, the steep terrain causes
chaotic wind shear distribution, with scattered high values in the central region, and
the maximum value increases to 4.31 m/s/10m. In summary, transitioning from
complex to flat terrain shifts the location of maximum VWS from mountainous areas
to flat valleys.

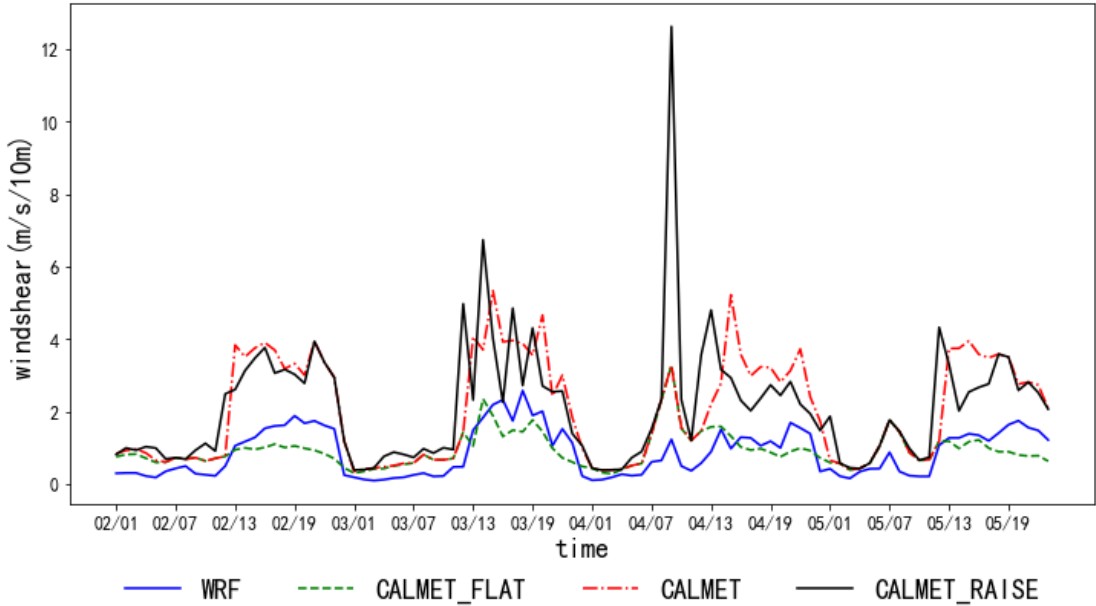


Figure 4:Time series of 10m-30m maximum VWS values of different
simulation experiments in the study area

This phenomenon is a typical result of valley winds, driven by the interaction
between terrain and solar radiation. During the day, sunlight heats the surface,
leading to differential heating rates between slopes and valleys due to their distinct
topographies. Slopes, receiving direct sunlight, warm up faster than valleys. At night,
the surface loses heat, particularly in valleys with good heat dissipation, resulting in
strong nighttime cooling effects. The temperature difference between slopes and
valleys during the day induces upslope airflow along the slopes. As the heated air
ascends, airflow forms over the valleys, as depicted in Figure 5(a) where maximum
VWS occurs near mountainous areas. At night, cold air flows downhill along the slopes,
forming downslope winds, which reverse the airflow pattern observed during the day.

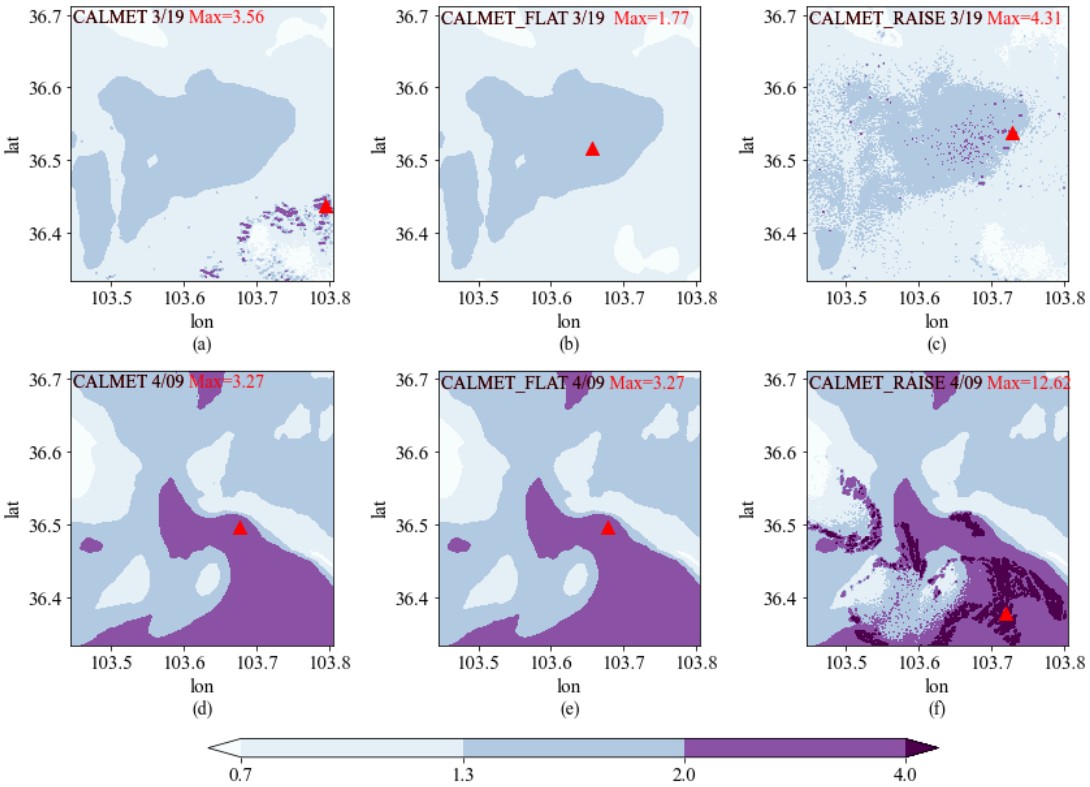


Figure 5: VWS distribution of 10m-30m.(a)-(c): 19:00 on July 3, 2022; (d)-(f):
09:00 on 4 July 2022; (a),(d):CALMET; (b),(e):CALMET_FLAT;
(c),(f):CALMET_RAISE.
The results indicate that CALMET model simulations of VWS are highly sensitive
to terrain: VWS values are generally lower in flat terrain compared to complex terrain,
and the influence of terrain on wind shear diminishes rapidly with height. In
extremely steep terrain, near-surface distribution appears chaotic, but VWS values
notably increase above the surface compared to complex terrain. Across the three
experiments, the absolute differences in VWS decrease with height, suggesting a
diminishing impact of terrain on CALMET model simulations of VWS with increasing
altitude.

To investigate extreme high values of VWS in the CALMET_RAISE experiment at
09:00 on July 4th, Figures 5(d)-(f) display the VWS distribution for all experiments at
this time, while Figure 6 presents wind vector maps for three hours for both CALEMT
and CALEMT_RAISE. The VWS distribution for CALEMT and CALEMT_FLAT is similar,
with a peak of 3.27m/s/10m in the central region. Compared to July 3rd at 19:00,
both experiments show extensive high-value areas in the southeast, with
CALMET_RAISE reaching an exceptional maximum of 12.62m/s/10m in the
southeastern valley area. Additionally, CALMET_RAISE exhibits large areas of
exceptionally high values compared to the other experiments.

In Figure 6, at 08:00 and 10:00 on July 4th, the prevailing wind direction in the
area is northeast. Both CALMET and CALEMT_RAISE show similar wind field
structures, transitioning from northeast to north as terrain slopes southward. When
airflow passes through the southern valley, mountain ranges create denser wind
vectors and increased speeds. However, at 09:00, a strong northwest airflow
converges with the northern airflow, forming a distinct "micro-front." The terrain
blocking induces diversion, deflection, and upward motion of the northwesterly wind,
creating extensive high-value VWS areas in the southeast. Compared to CALMET,
CALEMT_RAISE exhibits a more chaotic wind field due to increased terrain.

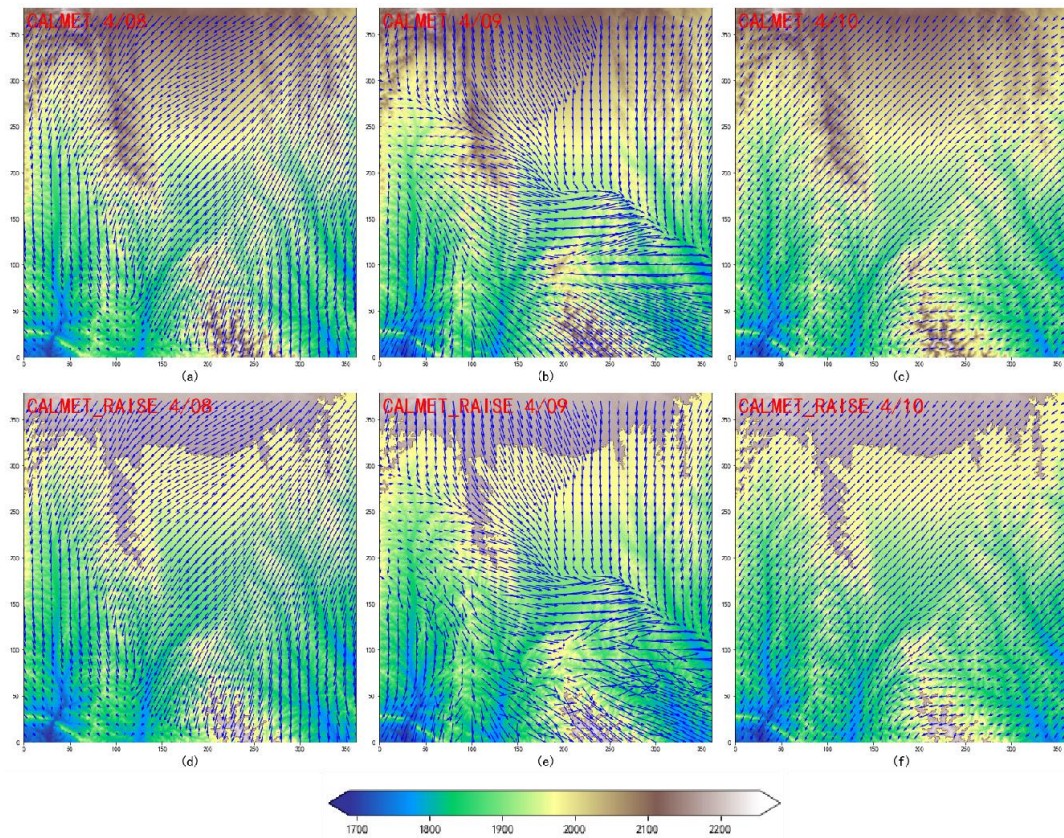

Figure 6. Topographic Elevation (unit: m) and Wind Vector Distribution. (a)-(c)
CALMET; (d)-(f) CALMET_RAISE; (a), (d) July 4th, 08:00; (b), (e) July 4th, 09:00; (c),
(f) July 4th, 10:00.
In conclusion, the widespread high-value VWS area observed at 09:00 on July 4th
resulted from a shift in wind direction to the northwest, encountering minimal
velocity reduction before reaching the tall terrain in the south, where the
mountainous obstruction led to diversion, deflection, and upward motion. The
anomalously high values in the CALEMT_RAISE experiment were attributed to the
elevation of the terrain, significantly intensifying the effects of diversion, deflection,
and upward motion. This suggests that terrain has a more pronounced impact on
CALMET-simulated wind shear during high wind speeds, while its influence is less
evident during low wind speeds. Therefore, heightened awareness of low-level wind
shear occurrence is warranted in complex terrain.

## 380 **4 Conclusion**

In order to investigate whether higher-resolution numerical models yield better
simulation results for low-level wind shear, this study focuses on a severe convective
weather event that occurred in the vicinity of Zhongchuan Airport on July 2, 2022.
The WRF/CALEMT coupled model is utilized to simulate the wind field, and the
influence of terrain variations on CALMET-simulated wind shear is explored. The
main conclusions are as follows:
(1) CALMET improves the simulation of near-surface winds, bringing them
closer to observed data than WRF, thereby facilitating more accurate modeling of
low-level wind shear.
(2) The diurnal variation of VWS shows a distinct pattern. CALMET exhibits
higher VWS compared to WRF, especially during the afternoon and evening. During
periods of intense convective activity, CALMET captures larger VWS fluctuations,
including higher peak values. CALMET's finer terrain features result in a VWS
distribution that better aligns with terrain effects, with VWS generally higher near
foothill areas compared to mountains, and a more pronounced decrease with altitude.
(3) Terrain sensitivity experiments show that during early morning and
morning hours, the maximum VWS of the three experiments were similar, occurring
in flat regions with minimal terrain influence. However, in the afternoon and evening,
CALMET_FLAT shows decreased maximum VWS values, while CALMET_RAISE
exhibits drastic fluctuations, with peak values near mountainous areas, indicating
significant terrain influence. Moreover, the impact of terrain on CALMET-simulated
VWS diminishes with altitude. These findings highlight the substantial influence of
terrain on CALMET, particularly during periods of high wind speeds.
(4) The occurrence of abnormally high VWS values in the simulations is
attributed to strong disturbances caused by tall terrain features: wind direction shifts
to northwest winds, encountering minimal reduction in wind speed before
encountering the tall terrain in the southern region. CALMET_RAISE elevates the
terrain from its original level, enhancing channeling, swirling, and updraft effects.
CALMET is a mature dynamic regional downscaling tool, and using other
numerical weather prediction models can also achieve the scale of CALMET. We chose
to use CALMET for the following reasons: from the perspective of operational
considerations, conducting research at the same scale requires lower computational
requirements and hardware needs for the CALMET model.

The research findings of this study are solely based on a short-term simulation
period of weather events in the Zhongchuan Airport area. However, this specific case
does not necessarily represent the overall wind shear situation at the airport, as it is
just one weather event with significant wind shear. And Obtaining radar wind profiler
data for the airport poses certain difficulties, we do not have Doppler lidar equipment
available. Direct observation of wind shear is challenging. We have made efforts to
obtain reanalysis data and site wind speed observations as much as possible. Due to
limited funding in the preliminary stages of our research, we could only start with
theoretical studies, and field experiments will be conducted once funding becomes
available.Our future work will expand to include longer simulation periods in more
airports and regions with complex terrain. This expansion aims to examine and
quantify the additional value provided by CALMET in simulating low-level wind shear.

*Competing Interests.* The corresponding author declares that all authors have no
competing interests.
**Acknowledgment**

This work was supported by the Joint Funds of the National Natural Science
Foundation of China (Grant No. U2342205), the Gansu Provincial Association of
Science and Technology Innovation Drive Promotion Project (Grant No.
GXH20230817-7), and the Key Natural Science Foundation of Gansu Province (Grant
No. 23JRRA1030). We extend our sincere appreciation to the funding agencies for
their support.

Additionally, we would like to express our gratitude to the Supercomputing
Center of Lanzhou University for their assistance.

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
