# Peer review of "Study on the influence of topography on wind shear-numerical simulation based on WRF-CALMET"

_EGUsphere, 2024_

## Author Response (AR1)

**Anonymous Referee #1, 06 Apr 2024:**

There are a couple of major limitations with the present study. One is the number of cases. Just one particular case is considered. Is it representative of windshear at that airport? Second is that the verification of the models is made with a times series of measured wind speed only. There are many discussions of vertical windshear in the manuscript. But they look like conjectures, and not substantiated by actual observations. Are there vertical wind profiles from radar wind profilers or Doppler LIDARs? Are there flight data from aircraft?

Moreover, the background of the windshear event is not discussed in detail. What is the synoptic and mesoscale patterns? Why there is windshear for the case? What is the nature of the windshear?

Also, CALMET as a dynamic downscaling tool is a bit out-dated. We can use numerical simulations by numerical weather prediction models at sub-kilometre skill or even sub-100 m scale for simulations. Of course they may be considered to be future studies, the limitations of CALMET model at least have to be spelt out.

Please provide a comprehensive review of terrain-induced windshear at other airports, and what observations and forecasting tools are used at these airports. With that as the background, discuss the novelty and scientific contributions of this manuscript.

**Response:**

Thank you very much for your questions and suggestions regarding our research.
1.One is the number of cases. Just one particular case is considered. Is it representative of windshear at that airport?

Response:we chose a particular case to study the impact of airport terrain on wind shear. However, this specific case does not necessarily represent the overall wind shear situation at the airport, as it is just one weather event with significant wind shear. Due to limitations in article length, we focused on studying this particular case in detail. We appreciate your suggestion, and we will address this limitation in the conclusions section.

2.Second is that the verification of the models is made with a times series of measured wind speed only.There are many discussions of vertical windshear in the manuscript.But they look like conjectures, and not substantiated by actual observations.Are there vertical wind profiles from radar wind profilers or Doppler LIDARs?Are there flight data from aircraft?

Response:Obtaining radar wind profiler data for the airport poses certain difficulties, and we do not have Doppler lidar equipment available. Direct observation of wind shear is challenging. We have made efforts to obtain reanalysis data and site wind speed observations as much as possible. Due to limited funding in the preliminary stages of our research, we could only start with theoretical studies, and field experiments will be conducted once funding becomes available. We appreciate your suggestion, and we will acknowledge this limitation in the conclusions.

3.The background of the windshear event is not discussed in detail. What is the synoptic and mesoscale patterns? Why there is windshear for the case? What is the nature of the windshear?

Response:we have discussed the background of the wind shear event in the supplement. The results indicate that during this period, a two-trough-two-ridge system in the East Asia region shifted eastward. This led to a deepening of the low-pressure trough at the location of the airport, with a significant strengthening of cold advection behind the ridge. Momentum descended from the upper atmosphere, and analysis of convective instability energy indicates significant convective activity in the airport area during this event. Therefore, this wind shear event was induced by terrain during convective weather in summer (Zhao, 2021).

4.CALMET as a dynamic downscaling tool is a bit out-dated. We can use numerical simulations by numerical weather prediction models at sub-kilometre skill or even sub-100 m scale for simulations. Of course they may be considered to be future studies, the limitations of CALMET model at least have to be spelt out.

Response: Sure, higher-resolution modeling modeling is a trend in model research. WRF can simulate scales down to sub-hundred-meter resolutions. From an operational perspective, the CALMET model only simulates wind fields, with computational demands and hardware requirements lower than the WRF model for simulations at the same scale. We believe that the CALMET model has research value in practical applications. Thank you for your suggestion, and we will address the limitations of CALMET in the conclusions.

5.Please provide a comprehensive review of terrain-induced windshear at other airports, and what observations and forecasting tools are used at these airports. With that as the background, discuss the novelty and scientific contributions of this manuscript.

Response: Thank you for your suggestion. We mentioned two cases of terrain-induced wind shear at Hong Kong International Airport using WRF and WRF-LES in the introduction of the preprint. However, we acknowledge that this

may not be sufficient, and we will include more background research to discuss the novelty and scientific contributions of the manuscript. Additionally, we plan to apply this method to study low-level wind shear at different airports in future research.

**Xiaohang Wen, 15 Apr 2024:**

Line 258, "4.2 Impact of Topography on Wind Shear Simulation" should be "3.2 Impact of Topography on Wind Shear Simulation".

Line 315 and 317, for "with a peak of 3.27m/s/10m ..." and "with CALMET_RAISE reaching an exceptional maximum of 12.62m/s/10m ...", are the units of vertical wind shear incorrect?

Line 329, do the x-axis and y-axis of Figure 6 consistent with the x-axis and y-axis of Figure 5? Are latitude and longitude used rather than 0, 50, 150, etc.?

**Response:**

Thank you very much for correcting the errors in my article and providing valuable feedback. I will promptly rectify issues 1 and 3.

Regarding issue 2, you raised a question about the unit of vertical wind shear. Typically, the unit of vertical wind shear is expressed in m/s. However, in my article, I opted to use m/s/10m to represent the wind shear magnitude between two 10-meter height layers. This unit signifies the rate of change of wind speed per 10 meters of vertical height. Such representation is common in meteorology and wind energy research, aiding in a better understanding of wind speed variations in the vertical direction, especially in contexts involving terrain and complex environmental influences.

---

## Author Response (AR2)

**Comments from the Associate Editor:**

In the manuscript, please carefully follow the comment of Reviewer #1: "Please provide a comprehensive review of terrain-induced windshear atother airports, and what observations and forecasting tools are used at these airports. With that as the background, discuss the novelty and scientific contributions of this manuscript." The response to this comment is quite short and superficial.

**Response:**

Thank you to the Associate Editor for the reminder. Indeed, in the previously submitted manuscript, my response to this comment from the reviewer was not thorough enough. In the revised manuscript, in the paragraphs from lines 41 to 95, I have included additional literature research, including numerical simulations of low-level wind shear at Nice Côte d'Azur Airport and commonly used numerical simulation methods for low-level wind shear. Additionally, at the end of the paragraph, I have demonstrated the novelty and scientific contribution of this manuscript.

**Comments from the Associate Editor:**

In the Supplement, please correct formatting issues (e.g., please justify all paragraphs) and add some relevant references.

**Response:**

Thank you for your reminder.I have corrected the formatting issues in the newly uploaded Supplement. In the fourth section, I have added subheadings to describe the purpose of each paragraph. Additionally, I have included some references in the Supplement.

---

## Author Response (AR3)

**Comments from the Associate Editor:**

In your manuscript, please use full first names for all authors. Although references are still based on initials, we will use full first names on the title page of your paper.

**Response:**

I have corrected the citation format in the manuscript, thank you very much for your suggestion.